# On $\varepsilon$-factorised bases and pure Feynman integrals

Hjalte Frellesvig[1] and Stefan Weinzierl[2]

**1** Niels Bohr International Academy, University of Copenhagen,
Blegdamsvej 17, 2100 København, Denmark
**2** PRISMA Cluster of Excellence, Institut für Physik,
Johannes Gutenberg-Universität Mainz, D - 55099 Mainz, Germany

## Abstract

We investigate $\varepsilon$-factorised differential equations, uniform transcendental weight and purity for Feynman integrals. We are in particular interested in Feynman integrals beyond the ones which evaluate to multiple polylogarithms. We show that a $\varepsilon$-factorised differential equation does not necessarily lead to Feynman integrals of uniform transcendental weight. We also point out that a proposed definition of purity works locally, but not globally.

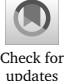 Check for updates

# 1 Introduction

The concepts of $\varepsilon$-factorised differential equations [1], uniform transcendental weight and purity, simple poles and constant leading singularities [2–4] play a crucial role in modern techniques for analytically computing Feynman integrals. These concepts are well understood for Feynman integrals which evaluate to multiple polylogarithms.

However, as soon as we leave this class of function not everything is as clear as we want it. This is already the case for the simplest Feynman integrals beyond the class of multiple polylogarithms, the ones which are associated to an elliptic curve. It is therefore timely and appropriate to clarify several issues. Although the main results of this paper are negative – we show that a certain basis does not have the uniform weight property and that a certain definition of purity does not apply globally to the simplest elliptic Feynman integral – we believe that exposing these subtleties is beneficial to progress in our understanding of Feynman integrals. The points which we discuss can be exemplified by the simplest elliptic Feynman integral, the two-loop sunrise integral with equal non-zero masses.

We start with $\varepsilon$-factorised differential equations. A $\varepsilon$-factorised differential equation together with boundary values at a given point allows for a systematic solution in terms of iterated integrals to any order in the dimensional regularisation parameter $\varepsilon$. But do these iterated integrals have additional nice properties like a definition of transcendental weight or integrands with simple poles only? In this paper we show that the general answer is no, but there might be bases of master integrals which have more of the nice properties than others.

This occurs already for the sunrise integral: We know two bases of master integrals, which put the associated differential equation into an $\varepsilon$-factorised form. The construction of either basis generalises to more complicated integrals, so it is worth examining the two bases in detail.

The first basis is constructed along the lines of an analysis of the maximal cut [5,6] and/or along the lines of prescriptive unitarity [7, 8]. Concretely this basis is constructed by the requirement that the period matrix on the maximal cut is proportional to the unit matrix [9]. For the sunrise integral we present a cleaned-up basis along these lines. Throughout this paper we denote this basis by $\vec{K}$.

The second basis is constructed from Picard-Fuchs operators and leads to a differential equation with modular forms [10]. For the sunrise integral we consider the basis given in [11]. This approach generalises nicely to more complicated Feynman integrals [12–18]. Throughout this paper we denote this basis by $\vec{J}$.

In this paper we work out the relation between the two bases. The first question we address is the following: Do these bases define master integrals of uniform weight? In principle, this requires a definition of transcendental weight for elliptic Feynman integrals. Let us first be agnostic to a full and complete definition of transcendental weight. We only make the minimal assumption that the definition of transcendental weight in the elliptic case should be compatible with the restriction of the kinematic space to a sub-space. With this assumption we may restrict to a point in kinematic space where the elliptic curve degenerates. The master integrals reduce to multiple polylogarithms, for which the definition of transcendental weight is unambiguous. Choosing this point as the boundary point for the integration of the differential equation forces the boundary constants (given by special values of multiple polylogarithms) to be of uniform weight (in the classical sense for multiple polylogarithms). In this way we may detect master integrals of non-uniform weight.

It turns out that basis $\vec{K}$ (constructed by the requirement that the period matrix on the maximal cut is proportional to the unit matrix) has boundary constants of non-uniform weight. Hence it is not a basis of uniform weight if we require that the notion of uniform weight is compatible with restrictions in the kinematic space.

The second question which we address in this paper is the relation between pure functions and logarithmic singularities. In order to answer this question we have to adopt a definition of purity for elliptic Feynman integrals. A generalisation of purity, which can be applied to the elliptic case, has been defined in ref. [19]: Functions which satisfy a differential equation without any homogeneous term are called unipotent. Unipotent functions, whose total differential involves only pure functions and one-forms with at most simple poles are called pure. Adopting this definition, we investigate if basis $\vec{J}$ (i.e. the modular form basis) for the sunrise integral is pure in this sense. We find that this is the case locally, but not globally. The argument which we present applies not only to the specific example of the equal mass sunrise integral, but to a wide range of elliptic Feynman integrals expressible in terms of the elliptic polylogarithms $\widetilde{\Gamma}$ [20]. We also present an argument that modifying the definition of purity by requiring that the above property holds only locally is too weak: It enlarges the function space too much.

This paper is organised as follows: In section 2 we start with a toy example, showing that an $\varepsilon$-factorised differential equation alone does not guarantee a solution of uniform weight. The boundary values need to be of uniform weight as well. The toy example is entirely within the class of multiple polylogarithms. In section 3 we introduce the standard example of an elliptic Feynman integral: the two-loop sunrise integral with equal non-zero masses. We introduce the notation which we will use in later sections of this paper.

In section 4 we investigate the first question: Are the known bases, which put the differential equation into an $\varepsilon$-factorised form also of uniform weight? In sub-section 4.1 we introduce three bases $\vec{I}$, $\vec{J}$ and $\vec{K}$ for the sunrise integral. The first one $\vec{I}$ is a pre-canonical basis and serves only in intermediate steps. The basis $\vec{J}$ is the one appearing in [11], while the basis $\vec{K}$ is the one appearing in [9]. The associated differential equations are given in sub-section 4.2. For the bases $\vec{J}$ and $\vec{K}$, the differential equations are in $\varepsilon$-factorised form. In sub-section 4.3 we discuss the period matrix on the maximal cut for the bases $\vec{J}$ and $\vec{K}$. By construction, the period matrix for the basis $\vec{K}$ is proportional to the unit matrix. In sub-section 4.4 we present the solutions for the master integrals for the bases $\vec{J}$ and $\vec{K}$. We then look at the values at $p^2 = 0$. At this point the elliptic curve degenerates and both solutions are given in terms of special values of multiple polylogarithms. We find that the basis $\vec{K}$ is not of uniform weight.

In section 5 we investigate the second question: What is the relation between purity and simple poles? We start in sub-section 5.1 with recapitulating the definition of purity from the literature. We then show in sub-section 5.2 that this definition does fit the modular form basis locally, but not globally. In sub-section 5.3 we demonstrate that our argument extends to Feynman integrals expressible in terms of elliptic polylogarithms $\widetilde{\Gamma}$. The problem is the behaviour at the finite cusps. However, modular transformations, which we discuss in sub-section 5.4, allow us to cover the kinematic space with coordinate charts such that in each coordinate chart the requirement from the definition of purity holds locally. Our conclusions are given in section 6. In appendix A we present the $q$-expansions of the modular forms and Eisenstein series appearing in the main text. In appendix B we give the boundary constants for the sunrise integral.

## 2 A toy example

We start with a simple toy example, showing that an $\varepsilon$-factorised differential equation alone does not guarantee a solution of uniform weight. The boundary values need to be of uniform weight as well.

Consider the two functions $F_1(x)$ and $F_2(x)$

$$
\begin{aligned}
F_1(x) &= e^{\varepsilon \ln(x)} \\
&= 1 + \varepsilon \ln(x) + \frac{1}{2}\varepsilon^2 (\ln(x))^2 + \mathcal{O}\left(\varepsilon^3\right), \\
F_2(x) &= (1+2\varepsilon) e^{\varepsilon \ln(x)} \\
&= 1 + \varepsilon\left[2 + \ln(x)\right] + \varepsilon^2\left[2\ln(x) + \frac{1}{2}(\ln(x))^2\right] + \mathcal{O}\left(\varepsilon^3\right).
\end{aligned}
\tag{1}
$$

$F_1(x)$ is of uniform weight (where we count algebraic numbers to be of weight zero, $\ln(x)$ to be of weight one, and $\varepsilon$ to be of weight minus one), while $F_2(x)$ is not. However, both function satisfy the $\varepsilon$-factorised differential equation

$$
\frac{d}{dx}F_i(x) = \frac{\varepsilon}{x}F_i(x), \qquad i \in \{1,2\}.
\tag{2}
$$

The general solution of eq. (2) as a power series in $\varepsilon$ reads

$$
F_i(x) = C_i^{(0)} + \left[C_i^{(1)} + C_i^{(0)} \ln(x)\right]\varepsilon + \left[C_i^{(2)} + C_i^{(1)} \ln(x) + \frac{1}{2}C_i^{(0)}(\ln(x))^2\right]\varepsilon^2 + \mathcal{O}\left(\varepsilon^3\right), \tag{3}
$$

with boundary values $C_i^{(j)}$, corresponding to the values of $F_i(x)$ at the point $x = 1$. For $F_1(x)$ the boundary values are

$$
C_1^{(0)} = 1, \qquad C_1^{(j)} = 0, \text{ for } j \geq 1.
\tag{4}
$$

For $F_2(x)$ the boundary values are

$$
C_2^{(0)} = 1, \qquad C_2^{(1)} = 2, \qquad C_2^{(j)} = 0, \text{ for } j \geq 2.
\tag{5}
$$

For a solution of uniform weight we must have that any non-zero boundary value $C_i^{(j)}$ is of weight $j$. This is the case for $F_1(x)$, but not for $F_2(x)$: The boundary value $C_2^{(1)}$ is of weight zero, for a solution of uniform weight it is supposed to be of weight one.

From this simple example we see that a $\varepsilon$-factorised differential equation alone does not guarantee a solution of uniform weight, we must in addition require that the boundary values $C_i^{(j)}\varepsilon^j$ are of uniform weight as well. This statement is of course obvious to experts in the field. We will use it in the following way: If we assume that a definition of transcendental weight beyond the polylogarihmic case is compatible with the restriction of the kinematic space to a sub-space, we may detect master integrals of non-uniform weight from their (non-uniform weight) boundary constants at a point where the master integrals reduce to values of multiple polylogarithms. For multiple polylogarithms the definition of transcendental weight is unambiguous.

## 3 Feynman integrals and elliptic curves

In this section we introduce the standard example of an elliptic Feynman integral: the two-loop sunrise integral with equal non-zero masses. This section also serves to set up the notation.

We consider the family of Feynman integrals

$$
I_{\nu_1 \nu_2 \nu_3}(D, x) = e^{2\varepsilon\gamma_E}\left(m^2\right)^{\nu_{123}-D} \int \frac{d^D k_1}{i\pi^{\frac{D}{2}}} \frac{d^D k_2}{i\pi^{\frac{D}{2}}} \frac{1}{\left(-q_1^2 + m^2\right)^{\nu_1}\left(-q_2^2 + m^2\right)^{\nu_2}\left(-q_3^2 + m^2\right)^{\nu_3}},
\tag{6}
$$

with $x = p^2/m^2$, $v_{123} = v_1 + v_2 + v_3$ and $q_1 = k_1$, $q_2 = k_2 - k_1$, $q_3 = -k_2 - p$. Below we will set $D = 2 - 2\varepsilon$.

The elliptic curve associated to this Feynman integral can be obtained from the maximal cut and is given by a quartic polynomial

$$P(u) = u(u+4)\left[u^2 + 2(1+x)u + (1-x)^2\right],\tag{7}$$

as

$$E : v^2 = P(u).\tag{8}$$

We denote the roots of the quartic polynomial $P(u)$ by

$$u_1 = -4, \quad u_2 = -\left(1 + \sqrt{x}\right)^2, \quad u_3 = -\left(1 - \sqrt{x}\right)^2, \quad u_4 = 0.\tag{9}$$

For $0 < x < 1$ the roots are real and ordered as

$$u_1 < u_2 < u_3 < u_4.\tag{10}$$

We set

$$\begin{aligned}
U_1 &= (u_3 - u_2)(u_4 - u_1) = 16\sqrt{x}, \\
U_2 &= (u_2 - u_1)(u_4 - u_3) = \left(1 - \sqrt{x}\right)^3\left(3 + \sqrt{x}\right), \\
U_3 &= (u_3 - u_1)(u_4 - u_2) = \left(1 + \sqrt{x}\right)^3\left(3 - \sqrt{x}\right).
\end{aligned}\tag{11}$$

We define the modulus and the complementary modulus of the elliptic curve $E$ by

$$k^2 = \frac{U_1}{U_3} = \frac{16\sqrt{x}}{\left(1 + \sqrt{x}\right)^3\left(3 - \sqrt{x}\right)}, \qquad \bar{k}^2 = 1 - k^2 = \frac{U_2}{U_3} = \frac{\left(1 - \sqrt{x}\right)^3\left(3 + \sqrt{x}\right)}{\left(1 + \sqrt{x}\right)^3\left(3 - \sqrt{x}\right)}.\tag{12}$$

Our standard choice for the periods and quasi-periods is

$$\begin{aligned}
\psi_1 &= \frac{4K(k)}{U_3^{\frac{1}{2}}}, & \psi_2 &= \frac{4iK(\bar{k})}{U_3^{\frac{1}{2}}}, \\
\phi_1 &= \frac{4[K(k) - E(k)]}{U_3^{\frac{1}{2}}}, & \phi_2 &= \frac{4iE(\bar{k})}{U_3^{\frac{1}{2}}}.
\end{aligned}\tag{13}$$

$K(x)$ and $E(x)$ denote the complete elliptic integral of the first kind and second kind, respectively:

$$K(x) = \int_0^1 \frac{dt}{\sqrt{(1-t^2)(1-x^2t^2)}}, \qquad E(x) = \int_0^1 dt\sqrt{\frac{1-x^2t^2}{1-t^2}}.\tag{14}$$

The geometric interpretation is as follows: For simplicity we assume that the roots $u_1$-$u_4$ are real and ordered as in eq. (10). The square root $v$ can be taken as a single-valued and continuous function on $\mathbb{C}\backslash([u_1, u_2] \cup [u_3, u_4])$

$$v = \sqrt{u - u_1}\sqrt{u - u_2}\sqrt{u_3 - u}\sqrt{u_4 - u},\tag{15}$$

where $\sqrt{x}$ denotes the standard square root with a branch cut along the negative real axis. For the ordering as in eq. (10) $v$ is positive for $u \in ]u_2, u_3[$. It is purely imaginary with positive

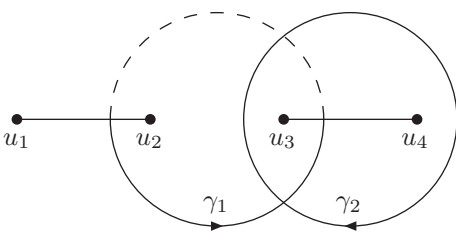

Figure 1: Branch cuts and cycles for the computation of the periods of an elliptic curve.

imaginary part just below the branch cut $[u_3, u_4]$. Let $\gamma_1$ and $\gamma_2$ be two cycles which generate the homology group $H_1(E, \mathbb{Z})$. This is shown in fig. 1. We choose $\gamma_1$ and $\gamma_2$ such that their intersection number is $(\gamma_1, \gamma_2) = +1$. Note that the intersection number is anti-symmetric: $(\gamma_2, \gamma_1) = -1$. The periods are alternatively given by

$$\psi_1 = \int_{\gamma_1} \frac{du}{v} = 2 \int_{u_2}^{u_3} \frac{du}{v}, \qquad \psi_2 = \int_{\gamma_2} \frac{du}{v} = 2 \int_{u_4}^{u_3} \frac{du}{v}. \tag{16}$$

In the last expression the square root is evaluated below the cut $[u_3, u_4]$. Similar formulae can be given for the quasi-periods.

The derivatives of the periods and quasi-periods are given for $i \in \{1, 2\}$ by

$$\frac{d}{dx}\psi_i = -\frac{1}{2}\psi_i \frac{d}{dx}(\ln U_2) + \frac{1}{2}\phi_i \frac{d}{dx}\left(\ln \frac{U_2}{U_1}\right),$$
$$\frac{d}{dx}\phi_i = -\frac{1}{2}\psi_i \frac{d}{dx}\left(\ln \frac{U_2}{U_3}\right) + \frac{1}{2}\phi_i \frac{d}{dx}\left(\ln \frac{U_2}{U_3^2}\right). \tag{17}$$

In particular we may use these relations to replace $\phi_i$ by $\frac{d\psi_i}{dx}$ or vice versa. Explicitly we have

$$3\left(1+\sqrt{x}\right)^2 \phi_i = 4\sqrt{x}\left(2+\sqrt{x}\right)\psi_i - 4x\left(1-\sqrt{x}\right)\left(3+\sqrt{x}\right)\frac{d}{dx}\psi_i. \tag{18}$$

Replacing $\phi_i$ by $\frac{d\psi_i}{dx}$ is often advantageous to eliminate the square root $\sqrt{x}$. In the following we will often write $\partial_x$ for $\frac{d}{dx}$. The Legendre relation reads

$$\psi_1 \phi_2 - \phi_1 \psi_2 = \frac{8\pi i}{\left(1+\sqrt{x}\right)^3 \left(3-\sqrt{x}\right)}. \tag{19}$$

We denote the Wronskian by

$$W = \psi_1 \partial_x \psi_2 - \psi_2 \partial_x \psi_1 = -\frac{6\pi i}{x\left(1-x\right)\left(9-x\right)}. \tag{20}$$

Finally, we set

$$\tau = \frac{\psi_2}{\psi_1}, \qquad q = e^{2\pi i \tau}. \tag{21}$$

We have

$$d\tau = \frac{W}{\psi_1^2} dx, \tag{22}$$

and

$$x = 9 \frac{\eta(\tau)^4 \eta(6\tau)^8}{\eta(3\tau)^4 \eta(2\tau)^8}, \tag{23}$$

where $\eta$ denotes Dedekind's eta-function. The first few terms read

$$x = 9q - 36q^2 + 90q^3 + \mathcal{O}\left(q^4\right). \tag{24}$$

## 4  Uniform weight and $\varepsilon$-factorised differential equations

In this section we investigate the question of uniform weight for bases of master integrals, which have $\varepsilon$-factorised differential equations. The two-loop sunrise integral with equal non-zero masses serves as an example.

### 4.1  Bases of master integrals

We consider three bases $\vec{I}, \vec{J}$ and $\vec{K}$ for the family of the sunrise integral. The first one, $\vec{I}$, is a basis without any additional properties and given by

$$\vec{I} = \begin{pmatrix} I_{110} \\ I_{111} \\ I_{211} \end{pmatrix}. \tag{25}$$

The latter two, $\vec{J}$ and $\vec{K}$, put the differential equation into an $\varepsilon$-form:

$$d\vec{J} = \varepsilon B \vec{J}, \qquad d\vec{K} = \varepsilon C \vec{K}, \tag{26}$$

where the $(3 \times 3)$-matrices $B$ and $C$ are independent of the dimensional regularisation parameter $\varepsilon$. The basis $\vec{J}$, appearing in [11, 21–23], is defined by

$$
\begin{aligned}
J_1 &= \varepsilon^2 I_{110}, \\
J_2 &= \varepsilon^2 \frac{\pi}{\psi_1} I_{111}, \\
J_3 &= \frac{\psi_1^2}{2\pi i \varepsilon W} \frac{d}{dx} J_2 + \frac{1}{24}\left(3x^2 - 10x - 9\right)\left(\frac{\psi_1}{\pi}\right)^2 J_2.
\end{aligned}
\tag{27}
$$

In terms of $I_{111}$ and $I_{211}$ the master integral $J_3$ is given by

$$
J_3 = \left[ -\frac{\varepsilon^2}{24}\left(x^2 - 30x + 45\right)\frac{\psi_1}{\pi} - \frac{\varepsilon}{4}\left(1 + \sqrt{x}\right)\left(3 - \sqrt{x}\right)\frac{\psi_1}{\pi} + \frac{\varepsilon}{16}\left(1 + \sqrt{x}\right)^3\left(3 - \sqrt{x}\right)\frac{\phi_1}{\pi} \right] I_{111}
$$

$$
+ \frac{\varepsilon}{4}(1 - x)(9 - x)\frac{\psi_1}{\pi} I_{211}. \tag{28}
$$

Note that the definition of the master integrals $\vec{J}$ involves only $\psi_1$ and $\phi_1$ (through $\frac{d}{dx}\psi_1$), but not $\psi_2$ nor $\phi_2$.

The basis $\vec{K}$, appearing in [9], is defined by

$$K_1 = \varepsilon^2 I_{110}, \tag{29}$$

$$K_2 = -\frac{\varepsilon(1 + 2\varepsilon)}{4\pi}\left(1 + \sqrt{x}\right)\left(3 - \sqrt{x}\right)\left[\psi_2 - \frac{1}{4}\left(1 + \sqrt{x}\right)^2 \phi_2\right] I_{111} + \frac{\varepsilon}{4\pi}(1 - x)(9 - x)\psi_2 I_{211},$$

$$K_3 = +\frac{\varepsilon(1 + 2\varepsilon)}{4\pi}\left(1 + \sqrt{x}\right)\left(3 - \sqrt{x}\right)\left[\psi_1 - \frac{1}{4}\left(1 + \sqrt{x}\right)^2 \phi_1\right] I_{111} - \frac{\varepsilon}{4\pi}(1 - x)(9 - x)\psi_1 I_{211}.$$

In the definition of the master integrals $\vec{K}$ all periods $\psi_1, \psi_2$ and all quasi-periods $\phi_1, \phi_2$ appear. The master integrals $K_2$ and $K_3$ are related by $\psi_2 \leftrightarrow \psi_1$, $\phi_2 \leftrightarrow \phi_1$ and an overall minus sign.

## 4.2 The differential equations

The differential equation in the basis $\vec{I}$ reads

$$d\vec{I} = A\vec{I}, \tag{30}$$

with

$$
A = \begin{pmatrix} 0 & 0 & 0 \\ 0 & -(1+2\varepsilon) & 3 \\ 0 & -\frac{1}{3}(1+2\varepsilon)(1+3\varepsilon) & 1+3\varepsilon \end{pmatrix} \frac{dx}{x}
$$
$$
+ \begin{pmatrix} 0 & 0 & 0 \\ 0 & 0 & 0 \\ \frac{\varepsilon^2}{4} & \frac{1}{4}(1+2\varepsilon)(1+3\varepsilon) & -(1+2\varepsilon) \end{pmatrix} \frac{dx}{x-1}
$$
$$
+ \begin{pmatrix} 0 & 0 & 0 \\ 0 & 0 & 0 \\ -\frac{\varepsilon^2}{4} & \frac{1}{12}(1+2\varepsilon)(1+3\varepsilon) & -(1+2\varepsilon) \end{pmatrix} \frac{dx}{x-9}. \tag{31}
$$

In this basis, the entries are rational dlog-forms. However, the differential equation is not in $\varepsilon$-form.

The differential equation in the basis $\vec{J}$ reads

$$d\vec{J} = \varepsilon B \vec{J}, \tag{32}$$

with

$$
B = \begin{pmatrix} 0 & 0 & 0 \\ 0 & \omega_2 & \omega_0 \\ \omega_3 & \omega_4 & \omega_2 \end{pmatrix}, \tag{33}
$$

and

$$
\begin{aligned}
\omega_0 &= 2\pi i \, d\tau = \frac{2\pi i W}{\psi_1^2} dx, \\
\omega_2 &= -f_2(\tau) \, (2\pi i) d\tau = \frac{dx}{2x} - \frac{dx}{x-1} - \frac{dx}{x-9}, \\
\omega_3 &= f_3(\tau) \, (2\pi i) d\tau = -\frac{1}{2}\frac{\psi_1}{\pi} dx, \\
\omega_4 &= f_4(\tau) \, (2\pi i) d\tau = \frac{(x+3)^4}{48x(x-1)(x-9)}\left(\frac{\psi_1}{\pi}\right)^2 dx.
\end{aligned} \tag{34}
$$

$f_2$, $f_3$ and $f_4$ are modular forms of $\Gamma_1(6)$. The minus sign in front of $f_2$ is convention. $\Gamma_1(N)$ is the subgroup of $\mathrm{SL}_2(\mathbb{Z})$ defined by

$$
\Gamma_1(N) = \left\{ \begin{pmatrix} a & b \\ c & d \end{pmatrix} \in \mathrm{SL}_2(\mathbb{Z}) : a,d \equiv 1 \bmod N,\ c \equiv 0 \bmod N \right\}. \tag{35}
$$

$\Gamma_1(N)$ is one of the standard congruence subgroups of $\mathrm{SL}_2(\mathbb{Z})$. A modular form $f(\tau)$ of a congruence subgroup $\Gamma$ is required to be holomorphic on the complex upper half-plane $\mathbb{H} = \{ \tau \in \mathbb{C} \mid \mathrm{Im}(\tau) > 0 \}$ and at the cusps. In addition it is required to transform under modular transformations $\gamma \in \Gamma$ as

$$
f\left(\frac{a\tau+b}{c\tau+d}\right) = (c\tau+d)^k \cdot f(\tau), \quad \text{for} \quad \gamma = \begin{pmatrix} a & b \\ c & d \end{pmatrix} \in \Gamma. \tag{36}
$$

Note that this transformation law is only required for $\gamma \in \Gamma$, but not for $\gamma \in \mathrm{SL}_2(\mathbb{Z})\backslash\Gamma$. Textbooks on modular forms and congruence subgroups are refs. [24–26], the appearance of modular forms in the context of this particular Feynman integral is discussed in more detail in refs. [10, 23].

In terms of the variable $x$ the modular forms $f_2, f_3$ and $f_4$ are given by

$$
f_2 = \frac{1}{24}\left(3x^2 - 10x - 9\right)\left(\frac{\psi_1}{\pi}\right)^2,
$$
$$
f_3 = -\frac{1}{24}x(x-1)(x-9)\left(\frac{\psi_1}{\pi}\right)^3,
$$
$$
f_4 = \frac{1}{576}(3+x)^4\left(\frac{\psi_1}{\pi}\right)^4. \tag{37}
$$

Their $q$-expansions are given in appendix A.

The differential equation in the basis $\vec{K}$ reads

$$
d\vec{K} = \varepsilon C \vec{K}, \tag{38}
$$

with

$$
C = \begin{pmatrix} 0 & 0 & 0 \\ C_{2,1} & C_{2,2} & C_{2,3} \\ C_{3,1} & C_{3,2} & C_{3,3} \end{pmatrix}, \tag{39}
$$

and

$$
C_{2,1} = -\frac{1}{2}\frac{\psi_2}{\pi}dx, \tag{40}
$$
$$
C_{2,2} = \frac{i\pi}{6}\left[(1+x)\frac{\psi_1}{\pi}\frac{\psi_2}{\pi} + \left(3x^2 - 10x - 9\right)\frac{\psi_2}{\pi}\frac{\partial_x\psi_1}{\pi} + 2x(x-1)(x-9)\frac{\partial_x\psi_1}{\pi}\frac{\partial_x\psi_2}{\pi}\right]dx,
$$
$$
C_{2,3} = \frac{i\pi}{6}\left[(1+x)\left(\frac{\psi_2}{\pi}\right)^2 + \left(3x^2 - 10x - 9\right)\frac{\psi_2}{\pi}\frac{\partial_x\psi_2}{\pi} + 2x(x-1)(x-9)\left(\frac{\partial_x\psi_2}{\pi}\right)^2\right]dx,
$$
$$
C_{3,1} = \frac{1}{2}\frac{\psi_1}{\pi}dx,
$$
$$
C_{3,2} = -\frac{i\pi}{6}\left[(1+x)\left(\frac{\psi_1}{\pi}\right)^2 + \left(3x^2 - 10x - 9\right)\frac{\psi_1}{\pi}\frac{\partial_x\psi_1}{\pi} + 2x(x-1)(x-9)\left(\frac{\partial_x\psi_1}{\pi}\right)^2\right]dx,
$$
$$
C_{3,3} = -\frac{i\pi}{6}\left[(1+x)\frac{\psi_1}{\pi}\frac{\psi_2}{\pi} + \left(3x^2 - 10x - 9\right)\frac{\psi_1}{\pi}\frac{\partial_x\psi_2}{\pi} + 2x(x-1)(x-9)\frac{\partial_x\psi_1}{\pi}\frac{\partial_x\psi_2}{\pi}\right]dx.
$$

### 4.3 Periods on the maximal cut

In this section we investigate the period matrices on the maximal cut of the sunrise integral. On the maximal cut of the sunrise integral only the last two master integrals are relevant (either $I_2, I_3$ or $J_2, J_3$ or $K_2, K_3$). The defining property for basis $\vec{K}$ is that the period matrix on the maximal cut is constant and proportional to the unit matrix.

We denote by

$$
\varphi_i^X, \qquad X \in \{I, J, K\}, \quad i \in \{1, 2, 3\}, \tag{41}
$$

the integrand of the master integral $X_i$ in the loop-by-loop Baikov representation [27]. In the loop-by-loop Baikov representations we have four integration variables $z_1 - z_4$, where $z_1 - z_3$ correspond to the three propagators and $z_4$ to an irreducible scalar product. Let $\mathcal{C}^{\mathrm{MaxCut}}$ be

the integration domain selecting the maximal cut, i.e. a small counter-clockwise circle around $z_1 = 0$, a small counter-clockwise circle around $z_2 = 0$ and a small counter-clockwise circle around $z_3 = 0$. We set $z_4 = u$ in accordance with the notation used in eq. (8). We denote by $\gamma_1$ and $\gamma_2$ the two cycles of the elliptic curve. They define the integration domain in the variable $u$. We define

$$\mathcal{C}_2 = \mathcal{C}^{\text{MaxCut}} \cup \gamma_1, \qquad \mathcal{C}_3 = \mathcal{C}^{\text{MaxCut}} \cup \gamma_2. \tag{42}$$

We consider the period matrix

$$P^X = \begin{pmatrix} \langle \varphi_2^X | \mathcal{C}_2 \rangle & \langle \varphi_2^X | \mathcal{C}_3 \rangle \\ \langle \varphi_3^X | \mathcal{C}_2 \rangle & \langle \varphi_3^X | \mathcal{C}_3 \rangle \end{pmatrix}. \tag{43}$$

In the $i$-th row of this matrix we then look at the leading term in the expansion in the dimensional regularisation parameter $\varepsilon$ for this row. We denote the order of the leading term of row $i$ by $j_{\min}(i)$. This defines a matrix $P^{X,\text{leading}}$ with entries

$$P_{ij}^{X,\text{leading}} = \text{coeff}\left(\langle \varphi_i^X | \mathcal{C}_j \rangle, \varepsilon^{j_{\min}(i)}\right) \cdot \varepsilon^{j_{\min}(i)}. \tag{44}$$

One finds

$$P^{I,\text{leading}} = -8i\pi \begin{pmatrix} \psi_1 & \psi_2 \\ \frac{\psi_1 - \frac{1}{4}(1+\sqrt{x})^2 \phi_1}{(1-\sqrt{x})(3+\sqrt{x})} & \frac{\psi_2 - \frac{1}{4}(1+\sqrt{x})^2 \phi_2}{(1-\sqrt{x})(3+\sqrt{x})} \end{pmatrix},$$

$$P^{J,\text{leading}} = 2i \begin{pmatrix} (2\pi i \varepsilon)^2 & (2\pi i \varepsilon)^2 \tau \\ 0 & -(2\pi i \varepsilon) \end{pmatrix},$$

$$P^{K,\text{leading}} = 4\pi\varepsilon \begin{pmatrix} 1 & 0 \\ 0 & 1 \end{pmatrix}. \tag{45}$$

We see that $P^{K,\text{leading}}$ is constant and proportional to the unit matrix. This is not surprising, since the basis $\vec{K}$ has been defined such that the whole period matrix on the maximal cut $P^K$ has that property, which of course implies that $P^{K,\text{leading}}$ has it as well.

Note that $P^{J,\text{leading}}$ can be written as

$$P^{J,\text{leading}} = 2i \begin{pmatrix} (2\pi i \varepsilon)^2 & 0 \\ 0 & -(2\pi i \varepsilon) \end{pmatrix} \begin{pmatrix} 1 & \tau \\ 0 & 1 \end{pmatrix}. \tag{46}$$

This is the decomposition of the period matrix $P^{J,\text{leading}}$ into a semi-simple matrix and an unipotent matrix [28, 29].

## 4.4 Solutions

In the basis $\vec{J}$ we may give a solution for the master integrals in terms of iterated integrals of modular forms.

Let $f_1(\tau), f_2(\tau), ..., f_n(\tau)$ be a set of modular forms. We define the $n$-fold iterated integral of these modular forms by

$$I(f_1, f_2, ..., f_n; \tau, \tau_0) = (2\pi i)^n \int_{\tau_0}^{\tau} d\tau_1 \int_{\tau_0}^{\tau_1} d\tau_2 \cdots \int_{\tau_0}^{\tau_{n-1}} d\tau_n \, f_1(\tau_1) f_2(\tau_2) \ldots f_n(\tau_n). \tag{47}$$

With $q = \exp(2\pi i \tau)$ we may equally well write

$$I(f_1, f_2, ..., f_n; \tau, \tau_0) = \int\limits_{q_0}^{q} \frac{dq_1}{q_1} \int\limits_{q_0}^{q_1} \frac{dq_2}{q_2} ... \int\limits_{q_0}^{q_{n-1}} \frac{dq_n}{q_n} f_1(\tau_1) f_2(\tau_2)...f_n(\tau_n), \qquad \tau_j = \frac{1}{2\pi i} \ln q_j.$$

(48)

It will be convenient to introduce a short-hand notation for repeated letters. We use the notation

$$\{f_i\}^j = \underbrace{f_i, f_i, ..., f_i}_{j},$$

(49)

to denote a sequence of $j$ letters $f_i$ and more generally

$$\left\{f_{i_1}, f_{i_2}, ..., f_{i_n}\right\}^j = \underbrace{f_{i_1}, f_{i_2}, ..., f_{i_n}, ......, f_{i_1}, f_{i_2}, ..., f_{i_n}}_{j \text{ copies of } f_{i_1}, f_{i_2}, ..., f_{i_n}},$$

(50)

to denote a sequence of $(j \cdot n)$ letters, consisting of $j$ copies of $f_{i_1}, f_{i_2}, ..., f_{i_n}$. For example

$$\{f_1, f_2\}^3 = f_1, f_2, f_1, f_2, f_1, f_2.$$

(51)

Our standard choice for the base point $\tau_0$ will be $\tau_0 = i\infty$, corresponding to $q_0 = 0$. This is unproblematic for modular forms which vanish at the cusp $\tau = i\infty$. In this case we have for a single integration

$$f = \sum_{j=1}^{\infty} a_j q^j \qquad \Rightarrow \qquad \int\limits_{0}^{q} \frac{dq_1}{q_1} f = \sum_{j=1}^{\infty} \frac{a_j}{j} q^j.$$

(52)

For modular forms which attain a finite value at the cusp $\tau = i\infty$ we employ the standard "trailing zero" or "tangential base point" regularisation [10, 30, 31]: We first take $q_0$ to have a small non-zero value. The integration will produce terms with $\ln(q_0)$. Let $R_{\ln(q_0)}$ be the operator, which removes all $\ln(q_0)$-terms. After these terms have been removed, we may take the limit $q_0 \to 0$. With a slight abuse of notation we set

$$I(f_1, f_2, ..., f_n; q) = \lim_{q_0 \to 0} R_{\ln(q_0)} \left[ \int\limits_{q_0}^{q} \frac{dq_1}{q_1} \int\limits_{q_0}^{q_1} \frac{dq_2}{q_2} ... \int\limits_{q_0}^{q_{n-1}} \frac{dq_n}{q_n} f_1(\tau_1) f_2(\tau_2)...f_n(\tau_n) \right].$$

(53)

We define the boundary constants $B_k$ for the sunrise integral $J_2$ by

$$\lim_{q \to 0} R_{\ln(q)} J_2 = e^{2 \sum\limits_{n=2}^{\infty} \frac{(-1)^n}{n} \zeta_n \varepsilon^n} \sum_{k=2}^{\infty} \varepsilon^k B_k.$$

(54)

The left-hand side corresponds to setting all iterated integrals to zero, including the ones which are proportional to powers of $\ln(q)$. The boundary values $B_k$ are collected in appendix B. Let us mention that the boundary values $B_k$ are of weight $k$. The right-hand side of eq. (54) is therefore of uniform weight.

We may express the master integrals in the basis $\vec{J}$ to all orders in the dimensional regularisation parameter in terms of iterated integrals of modular forms. We have

$$J_1 = e^{2\sum\limits_{n=2}^{\infty} \frac{(-1)^n}{n}\zeta_n \varepsilon^n},$$

$$J_2 = e^{-\varepsilon I(f_2;q)+2\sum\limits_{n=2}^{\infty} \frac{(-1)^n}{n}\zeta_n \varepsilon^n}\left\{\left[\sum_{j=0}^{\infty}\left(\varepsilon^{2j}I\left(\{1,f_4\}^j;q\right)-\frac{1}{2}\varepsilon^{2j+1}I\left(\{1,f_4\}^j,1;q\right)\right)\right]\sum_{k=2}^{\infty}\varepsilon^k B_k\right.$$
$$\left.+\sum_{j=0}^{\infty}\varepsilon^{j+2}\sum_{k=0}^{\lfloor\frac{j}{2}\rfloor}I\left(\{1,f_4\}^k,1,f_3,\{f_2\}^{j-2k};q\right)\right\},$$

$$J_3 = e^{-\varepsilon I(f_2;q)+2\sum\limits_{n=2}^{\infty} \frac{(-1)^n}{n}\zeta_n \varepsilon^n}\left\{\left[\sum_{j=0}^{\infty}\left(\varepsilon^{2j+1}I\left(\{f_4,1\}^j,f_4;q\right)-\frac{1}{2}\varepsilon^{2j}I\left(\{f_4,1\}^j;q\right)\right)\right]\sum_{k=2}^{\infty}\varepsilon^k B_k\right.$$
$$\left.+\sum_{j=0}^{\infty}\varepsilon^{j+1}\sum_{k=0}^{\lfloor\frac{j}{2}\rfloor}I\left(\{f_4,1\}^k,f_3,\{f_2\}^{j-2k};q\right)\right\}. \tag{55}$$

The expression for $J_2$ appeared already in [10], the expression for $J_3$ follows from (see eq. (27))

$$J_3 = \frac{1}{\varepsilon}\frac{1}{2\pi i}\frac{d}{d\tau}J_2 + f_2 J_2. \tag{56}$$

For the first few terms of $\varepsilon$-expansion we have

$$J_1 = 1 + \zeta_2\varepsilon^2 - \frac{2}{3}\zeta_3\varepsilon^3 + \frac{7}{10}\zeta_2^2\varepsilon^4 + \mathcal{O}\left(\varepsilon^5\right),$$

$$J_2 = \left[B_2 + I(1,f_3;q)\right]\varepsilon^2 + \left[B_3 - \frac{1}{2}B_2 I(1;q) - B_2 I(f_2;q) - I(1,f_2,f_3;q) - I(f_2,1,f_3;q)\right]\varepsilon^3$$
$$+\left[B_4 + \zeta_2 B_2 - \frac{1}{2}B_3 I(1;q) - B_3 I(f_2;q) + \frac{1}{2}B_2 I(1,f_2;q) + \frac{1}{2}B_2 I(f_2,1;q)\right.$$
$$+B_2 I\left(1,f_4;q\right) + B_2 I(f_2,f_2;q) + \zeta_2 I(1,f_3;q) + I(1,f_2,f_2,f_3;q) + I(f_2,f_2,1,f_3;q)$$
$$\left.+I\left(1,f_4,1,f_3;q\right) + I(f_2,1,f_2,f_3;q)\right]\varepsilon^4 + \mathcal{O}\left(\varepsilon^5\right),$$

$$J_3 = \varepsilon I(f_3;q) + \left[-\frac{1}{2}B_2 - I(f_2,f_3;q)\right]\varepsilon^2 + \left[-\frac{1}{2}B_3 + \frac{1}{2}B_2 I(f_2;q) + B_2 I\left(f_4;q\right)\right.$$
$$\left.+\zeta_2 I(f_3;q) + I(f_2,f_2,f_3;q) + I\left(f_4,1,f_3;q\right)\right]\varepsilon^3 + \left[-\frac{1}{2}B_4 - \frac{1}{2}\zeta_2 B_2 + \frac{1}{2}B_3 I(f_2;q)\right.$$
$$+B_3 I\left(f_4;q\right) - \frac{2}{3}\zeta_3 I(f_3;q) - B_2 I\left(f_4,f_2;q\right) - B_2 I\left(f_2,f_4;q\right) - \frac{1}{2}B_2 I(f_2,f_2;q)$$
$$-\frac{1}{2}B_2 I\left(f_4,1;q\right) - \zeta_2 I(f_2,f_3;q) - I(f_2,f_2,f_2,f_3;q) - I\left(f_4,f_2,1,f_3;q\right)$$
$$\left.-I\left(f_2,f_4,1,f_3;q\right) - I\left(f_4,1,f_2,f_3;q\right)\right]\varepsilon^4 + \mathcal{O}\left(\varepsilon^5\right). \tag{57}$$

Let us also summarise the boundary values: From eq. (54) and eq. (55) we obtain

$$\lim_{q \to 0} R_{\ln(q)} J_1 = e^{2 \sum_{n=2}^{\infty} \frac{(-1)^n}{n} \zeta_n \varepsilon^n},$$

$$\lim_{q \to 0} R_{\ln(q)} J_2 = e^{2 \sum_{n=2}^{\infty} \frac{(-1)^n}{n} \zeta_n \varepsilon^n} \sum_{k=2}^{\infty} \varepsilon^k B_k,$$

$$\lim_{q \to 0} R_{\ln(q)} J_3 = -\frac{1}{2} e^{2 \sum_{n=2}^{\infty} \frac{(-1)^n}{n} \zeta_n \varepsilon^n} \sum_{k=2}^{\infty} \varepsilon^k B_k. \tag{58}$$

In all three cases the right-hand sides are of uniform weight.

Given a solution in the basis $\vec{J}$, we easily obtain a solution in the basis $\vec{K}$. The two bases are related by

$$\vec{K} = U \vec{J}, \tag{59}$$

with

$$U = \begin{pmatrix} 1 & 0 & 0 \\ 0 & -\frac{(1+2\varepsilon)}{2\pi i \varepsilon} - g_2 \cdot \tau & \tau \\ 0 & g_2 & -1 \end{pmatrix}, \tag{60}$$

and

$$g_2 = \frac{1}{24} \left[ (3x^2 - 10x - 9) \frac{\psi_1}{\pi} + 4x(1-x)(9-x) \frac{\partial_x \psi_1}{\pi} \right] \frac{\psi_1}{\pi}. \tag{61}$$

In the modular variable $\tau$ the quantity $g_2$ is given by

$$g_2 = f_2 + 2 \frac{\pi}{\psi_1} \frac{1}{2\pi i} \frac{d}{d\tau} \frac{\psi_1}{\pi}$$
$$= 4 \left( 3b_1^2 - 3b_1 b_2 - 6b_2^2 - e_2 \right). \tag{62}$$

The modular forms $b_1$ and $b_2$ and the quasi-modular form $e_2$ are defined in appendix A. The quantity $g_2$ is a quasi-modular form of modular weight 2 and depth 1. For $\gamma \in \Gamma_1(6)$ the quantity $g_2$ transforms as

$$(g_2|_2 \gamma)(\tau) = g_2(\tau) + \frac{2}{2\pi i} \frac{c}{c\tau + d}, \qquad \gamma = \begin{pmatrix} a & b \\ c & d \end{pmatrix}, \tag{63}$$

where the operator $|_k \gamma$ is defined by

$$(f|_k \gamma)(\tau) = (c\tau + d)^{-k} \cdot f(\gamma(\tau)), \qquad \gamma(\tau) = \frac{a\tau + b}{c\tau + d}. \tag{64}$$

For the first few terms of $\varepsilon$-expansion we have

$$K_2 = \frac{1}{2\pi i} \left[ -B_2 + I(f_3, 1; q) \right] \varepsilon + \frac{1}{2\pi i} \left[ -B_3 - 2B_2 + B_2 I(f_2; q) - I(f_2, f_3, 1; q) - 2I(1, f_3; q) \right.$$
$$\left. -2g_2 I(1, 1, f_3; q) - g_2 I(1, f_3, 1; q) - g_2 B_2 I(1; q) \right] \varepsilon^2 + \mathcal{O}(\varepsilon^3),$$

$$K_3 = -I(f_3; q) \varepsilon + \left[ \frac{1}{2} B_2 + I(f_2, f_3; q) + g_2 B_2 + g_2 I(1, f_3; q) \right] \varepsilon^2 + \mathcal{O}(\varepsilon^3). \tag{65}$$

Let us look at the boundary values of $K_2$

$$\lim_{q\to 0} R_{\ln(q)} K_2 = -\frac{B_2}{2\pi i}\varepsilon - \frac{(B_3 + 2B_2)}{2\pi i}\varepsilon^2 + \mathcal{O}\left(\varepsilon^3\right). \tag{66}$$

The term

$$-\frac{2B_2}{2\pi i}\varepsilon^2, \tag{67}$$

is of weight minus one and spoils the uniform weight property. Hence we conclude that the basis $\vec{K}$ is not of uniform weight if we require that the notion of uniform weight is compatible with restrictions in the kinematic space.

## 5 Purity and simple poles

In this section we address the second main question of this paper: The relation between purity and simple poles in the elliptic case.

### 5.1 Definition of pure functions in the literature

We recapitulate the definitions of unipotent and pure function as given in ref. [19]:

**Definition 1.** *A function is called unipotent, if it satisfies a differential equation without a homogeneous term.*

To give an example, the functions $\ln(x)$ and $\text{Li}_2(x)$ are unipotent

$$\frac{d}{dx}\ln(x) = \frac{1}{x}, \qquad \frac{d}{dx}\text{Li}_2(x) = -\frac{1}{x}\ln(1-x), \tag{68}$$

while $e^x$ is not

$$\frac{d}{dx}e^x = e^x. \tag{69}$$

**Definition 2.** *Unipotent functions, whose total differential involves only pure functions and one-forms with at most simple poles are called pure.*

The standard example are multiple polylogarithms, whose total differential is given by

$$dG(z_1,\ldots,z_r;y) = \sum_{j=1}^{r} G(z_1,\ldots,\hat{z}_j,\ldots,z_r;y)\left[d\ln\left(z_{j-1}-z_j\right) - d\ln\left(z_{j+1}-z_j\right)\right], \tag{70}$$

where we set $z_0 = y$ and $z_{r+1} = 0$. A hat indicates that the corresponding variable is omitted. In addition one uses the convention that for $z_{j+1} = z_j$ the one-form $d\ln(z_{j+1}-z_j)$ equals zero. Clearly, the one forms

$$d\ln\left(z_{j+1}-z_j\right) = \frac{dz_{j+1}-dz_j}{z_{j+1}-z_j}, \tag{71}$$

have only simple poles.

## 5.2 Iterated integrals of modular forms

Let us now look at iterated integrals of modular forms, as defined in eq. (47). It is clear that these iterated integrals are unipotent functions, as differentiation removes one integration. We investigate the order of the poles of the total differential.

We denote by

$$\mathbb{H} = \{\, \tau \in \mathbb{C} \mid \mathrm{Im}(\tau) > 0 \,\}\,, \tag{72}$$

the complex upper half-plane and by

$$\overline{\mathbb{H}} = \mathbb{H} \cup \{i\infty\} \cup \mathbb{Q}\,, \tag{73}$$

the extended complex upper half-plane. Under the map $q = \exp(2\pi i \tau)$ the complex upper half-plane $\mathbb{H}$ is mapped to the punctured open disk

$$D = \{\, q \in \mathbb{C} \mid 0 < |q| < 1 \,\}\,, \tag{74}$$

and $\overline{\mathbb{H}}$ is mapped to

$$\overline{D} = D \cup \{0\} \cup \left\{\, e^{2\pi i r} \mid r \in \mathbb{Q} \,\right\}\,. \tag{75}$$

Let $f_k(\tau)$ be a modular form of weight $k$ for a congruence subgroup $\Gamma$ and

$$\omega_k^{\mathrm{modular}} = 2\pi i \, f_k(\tau)\, d\tau\,. \tag{76}$$

For simplicity we assume that $\begin{pmatrix} 1 & 1 \\ 0 & 1 \end{pmatrix} \in \Gamma$. In this case $f_k$ has the $q$-expansion [25]

$$f_k = \sum_{n=0}^{\infty} a_n q^n\,. \tag{77}$$

(In the general case $f_k$ will have an expansion in $q^{\frac{1}{N'}}$, where $N'$ is the smallest positive integer such that $f_k(\tau + N') = f_k(\tau)$. The general case is only from a notational perspective more elaborate.) In addition we will always assume that modular forms are normalised such that the coefficients of their $q$-expansion are algebraic numbers. This is a convenient convention.[1] We view $\omega_k^{\mathrm{modular}}$ as a differential one-form on $\overline{D}$. In the variable $q$ we have

$$\omega_k^{\mathrm{modular}} = \sum_{n=0}^{\infty} a_n q^{n-1} dq\,. \tag{78}$$

This shows immediately that $\omega_k^{\mathrm{modular}}$ is holomorphic on $D$ and has a simple pole at $q = 0$ if $a_0 \neq 0$. Thus, in a neighbourhood of $q = 0$ the differential one-form $\omega_k^{\mathrm{modular}}$ has at most simple poles.

Let us now discuss if this extends globally to $\overline{D}$. The answer will be no. We have to look at the other cusps. We investigate the behaviour at

$$q_0 = e^{2\pi i \left(-\frac{d}{c}\right)}\,, \qquad c, d \in \mathbb{Z}, \quad c \neq 0\,. \tag{79}$$

---

[1] For the modular group $\mathrm{SL}_2(\mathbb{Z})$ this implies that we work with the Eisenstein series $\frac{1}{(2\pi i)^k} \sum_{\substack{(n_1, n_2) \in \mathbb{Z}^2 \setminus (0,0)}}^{e} \frac{1}{(n_1 + n_2 \tau)^k}$, which have rational $q$-expansion coefficients instead of $\sum_{\substack{(n_1, n_2) \in \mathbb{Z}^2 \setminus (0,0)}}^{e} \frac{1}{(n_1 + n_2 \tau)^k}$, where every $q$-expansion coefficient is a rational multiple of $(2\pi i)^k$.

We may derive the behaviour of $\omega_k^{\text{modular}}$ at $q_0$ from the modular properties of $f_k$. We consider the modular transformation

$$\gamma = \begin{pmatrix} a & b \\ c & d \end{pmatrix} \in \text{SL}_2(\mathbb{Z}), \qquad \gamma^{-1} = \begin{pmatrix} d & -b \\ -c & a \end{pmatrix}, \tag{80}$$

and set

$$\tau' = \gamma(\tau) = \frac{a\tau + b}{c\tau + d}, \qquad q' = e^{2\pi i \tau'}. \tag{81}$$

This maps $\tau = -\frac{d}{c}$ to $\tau' = i\infty$ and $q_0$ to $q_0' = 0$. For the automorphic factor we have

$$c\tau + d = \frac{c}{2\pi i} \frac{(q - q_0)}{q_0} + \mathcal{O}\left((q - q_0)^2\right). \tag{82}$$

$(f_k|_k \gamma^{-1})(\tau')$ has again a $q'$-expansion as in eq. (77)

$$\left(f_k|_k \gamma^{-1}\right)\left(\tau'\right) = \sum_{n=0}^{\infty} a_n' \left(q'\right)^n. \tag{83}$$

If $f_k$ is a modular form for the congruence subgroup $\Gamma$ and $\gamma \in \Gamma$ we have $a_n' = a_n$, otherwise the coefficients need not be the same. Usually we are interested in the cusps not equivalent to $\tau = i\infty$, this implies $\gamma \in \text{SL}_2(\mathbb{Z}) \backslash \Gamma$. For $a_0' \neq 0$ we have

$$\omega_k^{\text{modular}} = a_0' \left(\frac{c}{2\pi i}\right)^{-k} q_0^{k-1} \frac{dq}{(q - q_0)^k} + \mathcal{O}\left((q - q_0)^{-k+1}\right). \tag{84}$$

Thus we see that whenever $f_k$ is non-vanishing at the cusp $\tau_0 = -\frac{d}{c}$, the differential one-form $\omega_k^{\text{modular}}$ has a pole of order $k$ in the variable $q$ at $q = q_0$. Globally, $\omega_k^{\text{modular}}$ has poles up to order $k$ on $\overline{D}$.

These poles do occur. Consider for example the modular form $f_3$ from eq. (37). It is a modular form of modular weight 3 for the congruence subgroup $\Gamma_1(6)$. The space of modular forms $\mathcal{M}_3(\Gamma_1(6))$ of modular weight 3 for $\Gamma_1(6)$ is four-dimensional and consists solely of the Eisenstein subspace.[2] There are no cusp forms in this space. Hence, $f_3 \in \mathcal{M}_3(\Gamma_1(6))$ is an Eisenstein series. From the $q$-expansion of eq. (A.5) one sees that $f_3$ vanishes at the cusp $\tau = i\infty$. As $f_3$ is not a cusp form, there must be a cusp $\tau \in \mathbb{Q}$, where $f_3$ is not vanishing. Hence, $\omega_3 = 2\pi i f_3(\tau) d\tau$ has a pole of order three in the variable $q$ there.

## 5.3 Elliptic polylogarithms

The discussion of the previous sub-section is not restricted to iterated integrals of modular forms and carries over to elliptic polylogarithms $\widetilde{\Gamma}$.

Let $g^{(k)}(z, \tau)$ denote the coefficients of the Kronecker function and set

$$\omega_k^{\text{Kronecker}}(z, \tau) = (2\pi i)^{2-k} \left[ g^{(k-1)}(z, \tau) \, dz + (k-1) g^{(k)}(z, \tau) \frac{d\tau}{2\pi i} \right]. \tag{85}$$

We may view $\omega_k^{\text{Kronecker}}$ as a differential one-form on the two-dimensional moduli space $\mathcal{M}_{1,2}$. Coordinates on this moduli space are $(z, \tau)$. The elliptic polylogarithms $\widetilde{\Gamma}$ are iterated integrals of $\omega_k^{\text{Kronecker}}(z - c_j, \tau)$ along $z$ at constant $\tau$. It is known that the functions $g^{(k)}(z, \tau)$ have at

---

[2]The computer algebra system `Sage` can be used to obtain a basis of $\mathcal{M}_3(\Gamma_1(6))$. The underlying mathematics is explained in refs. [24, 25].

most simple poles in $z$ and when restricted to $\tau = $ const the elliptic polylogarithms $\widetilde{\Gamma}$ are pure functions in the sense of definition 2. However in the applications towards Feynman integrals it is usually the case that the assumption $\tau = $ const is not justified. A variation of the kinematic variables of the Feynman integral will imply a variation of $\tau$ and we have to consider the $\tau$-dependence as well. For the argument we want to make it is sufficient to restrict to $z = a + b\tau$ with $a, b \in \mathbb{Q}$ and $k \geq 2$. In this case the differential one-forms $\omega_k^{\text{Kronecker}}$ reduce to the form of $\omega_k^{\text{modular}}$ [28] and the argument from the previous sub-section applies: In this case the differential one-forms $\omega_k^{\text{Kronecker}}$ may have poles up to order $k$ in the variable $q$ (or $\tau$).

## 5.4 Modular transformations

We have seen that locally in the coordinate chart $D \cup \{0\}$ the basis $\vec{J}$ satisfies the criteria of definition 2. This coordinate chart includes the point $x = 0$. Let us now investigate the global picture. For the sunrise integral we have four singular points $x \in \{0, 1, 9, \infty\}$ and we may cover the kinematic space with four charts, such that each chart includes exactly one singular point [21, 22, 32]. Below we will follow the notation of ref. [22].

In each chart we may construct a basis, which satisfies the criteria of definition 2 locally. In different charts we will have different coordinates $\tau$ and $\tau'$, but also different bases of master integrals $\vec{J}$ and $\vec{J}'$. The coordinates $\tau$ and $\tau'$ will be related by a modular transformation. The modular transformation induces also the transformation between $\vec{J}$ and $\vec{J}'$.

Let us discuss the behaviour near the cusp $\tau_0 = -\frac{d}{c}$. The modular transformation $\gamma$ defined in eq. (80) maps $\tau_0 = -\frac{d}{c}$ to $\tau'_0 = i\infty$. Let $f_k$ be a modular form for a congruence subgroup $\Gamma$. Then by definition $f_k(\tau)$ is holomorphic on $\mathbb{H}$ and $(f_k|_k\gamma^{-1})(\tau')$ has a $q'$-expansion as in eq. (83) for any $\gamma \in \mathrm{SL}_2(\mathbb{Z})$. This suggest to change in a neighbourhood of $\tau = -\frac{d}{c}$ coordinates from $q$ to $q'$. The differential one-form

$$2\pi i \left(f_k|_k\gamma^{-1}\right)\left(\tau'\right) d\tau', \tag{86}$$

has then a simple pole at $q' = 0$ (corresponding to $\tau = -\frac{d}{c}$). However, $\omega_k^{\text{modular}}$ as defined in eq. (76) does not transform under this coordinate change into eq. (86). Instead we find

$$\omega_k^{\text{modular}} = \left(-c\tau' + a\right)^{k-2} \cdot 2\pi i \left(f_k|_k\gamma^{-1}\right)\left(\tau'\right) d\tau'. \tag{87}$$

$(-c\tau' + a)$ is the automorphic factor for $\gamma^{-1}$. For $k \neq 2$ this factor spoils that iterated integrals of modular forms transform under modular transformations into iterated integrals of modular forms. However, elliptic Feynman integrals transform nicely: Let us consider for

$$\gamma(\tau) = \frac{a\tau + b}{c\tau + d}, \qquad \gamma \in \mathrm{SL}_2(\mathbb{Z}), \tag{88}$$

the combined transformation

$$\vec{J}' = \begin{pmatrix} 1 & 0 & 0 \\ 0 & \frac{1}{c\tau+d} & 0 \\ 0 & -\frac{c}{2\pi i \varepsilon} & c\tau+d \end{pmatrix} \vec{J},$$

$$\tau' = \frac{a\tau + b}{c\tau + d}. \tag{89}$$

One obtains

$$d\vec{J}' = \varepsilon B' \vec{J}', \tag{90}$$

with

$$B' = 2\pi i \begin{pmatrix} 0 & 0 & 0 \\ 0 & -(f_2|_2\gamma^{-1})(\tau') & 1 \\ (f_3|_3\gamma^{-1})(\tau') & (f_4|_4\gamma^{-1})(\tau') & -(f_2|_2\gamma^{-1})(\tau') \end{pmatrix} d\tau'. \tag{91}$$

As $\Gamma(6)$ is a subgroup of $\Gamma_1(6)$ we have $\mathcal{M}_k(\Gamma_1(6)) \subset \mathcal{M}_k(\Gamma(6))$ and as $\Gamma(6)$ is a normal subgroup of $\mathrm{SL}_2(\mathbb{Z})$ it follows that

$$f_k|_k\gamma^{-1} \in \mathcal{M}_k(\Gamma(6)). \tag{92}$$

We see that the entries of $B'$ are again differential one-forms of the form as in eq. (76). We may express $\vec{J}'$ again in terms of iterated integrals of modular forms, this time in the variable $q'$. It can be shown that the boundary constants are again of uniform weight. Hence it follows that in the coordinate chart with coordinate $q'$ (or $\tau'$) the basis $\vec{J}'$ satisfies the criteria of definition 2 locally.

Although the kinematic space of the sunrise family can be covered with four charts such that in each chart the criteria of definition 2 holds locally, we do not advocate to define purity by requiring that the kinematic space can be covered with local charts, such that in each chart the criteria of definition 2 holds locally. The reason is given by the following counterexample: Consider the functions $f_j(x)$ defined by the generating series

$$\sum_{j=0}^{\infty} f_j(x)\,\varepsilon^j = x^{-\varepsilon} e^{\frac{\varepsilon}{1-x}} \tag{93}$$

$$= 1 + \left[\frac{1}{1-x} - \ln(x)\right]\varepsilon + \frac{1}{2}\left[\frac{1}{(1-x)^2} - \frac{2\ln(x)}{1-x} + \ln^2(x)\right]\varepsilon^2 + \mathcal{O}\left(\varepsilon^3\right).$$

Clearly, we would not like to call the functions $f_j$ for $j \geq 1$ pure. The functions $f_j$ satisfy the differential equations

$$df_j = \left[-\frac{dx}{x} + \frac{dx}{(1-x)^2}\right] f_{j-1}. \tag{94}$$

The functions $f_j$ are clearly unipotent. In a neighbourhood of $x = 0$ the differential has a simple pole at $x = 0$. In a neighbourhood of $x = 1$ the change of variables

$$x' = e^{\frac{1}{1-x}}, \tag{95}$$

transforms the one-form with a double pole into a one-form with a simple pole:

$$\frac{dx}{(1-x)^2} = \frac{dx'}{x'}. \tag{96}$$

Thus we see that modifying the definition of purity by requiring that definition 2 holds only locally is too weak: It enlarges the function space too much.

Let us however point out a difference between the sunrise integral and the counterexample: In the latter case the transformation in eq. (95) is a general ad-hoc coordinate transformation, whereas in eq. (89) we only consider the smaller set of transformations induced by modular transformations.

## 6 Conclusions

For Feynman integrals which evaluate to multiple polylogarithms we have a clear understanding of uniform weight and purity: These are Feynman integrals, whose term of order $j$ in the $\varepsilon$-expansion is pure of transcendental weight $j$. We are interested in extending this concept to Feynman integrals beyond the ones which evaluate to multiple polylogarithms.

This is non-trivial and in this paper we discussed some subtleties: We showed that an $\varepsilon$-factorised differential equation alone does not necessarily lead to a solution of uniform transcendental weight. The boundary values have to be of uniform transcendental weight as well. This applies in particular to a basis constructed by the requirement that the period matrix on the maximal cut is proportional to the unit matrix. The argument we presented is agnostic to the exact definition of weight beyond the case of multiple polylogarithms, we only assumed that the definition of transcendental weight in the general case is compatible with the restriction of the kinematic space to a sub-space.

In the second part of the paper we adopted a particular definition of purity from the literature. We showed that this definition works only locally – but not globally – for a particular basis of the two-loop equal mass sunrise integral. Of course, it might well be that this particular basis is not the optimal one, but another possibility is that the definition of purity needs a more refined definition. The modular transformation properties, which we discussed in section 5.4, point towards a possible modification.

We believe that the detailed analysis we carried out in this paper will be helpful for a definition of purity which not only includes the elliptic case, but also Feynman integrals related to Calabi-Yau geometries.

## Acknowledgments

S.W. thanks the Niels Bohr Institute for hospitality and H.F. thanks the Mainz Institute of Theoretical Physics for hospitality.

**Funding information**  H.F. is partially supported by a Carlsberg Foundation Reintegration Fellowship, and has received funding from the European Union's Horizon 2020 research and innovation program under the Marie Sklodowska-Curie grant agreement No. 847523 "INTER-ACTIONS".

## A  Modular forms

In this appendix we give the $q$-expansions of the modular forms $f_2$, $f_3$ and $f_4$, appearing in eq. (34) and the $q$-expansions of $\psi_1$ (which is a modular form of modular weight 1). In addition, we define the Eisenstein series $e_2$, which appears in eq. (62).

We start by introducing a basis $\{b_1, b_2\}$ for the modular forms of modular weight 1 for the Eisenstein subspace $\mathcal{E}_1(\Gamma_1(6))$:

$$b_1 = E_1(\tau; \chi_1, \chi_{-3}), \qquad b_2 = E_1(2\tau; \chi_1, \chi_{-3}), \qquad (A.1)$$

where $\chi_1$ and $\chi_{-3}$ denote primitive Dirichlet characters with conductors 1 and 3, respectively. In terms of the coefficients $g^{(k)}(z, \tau)$ of the Kronecker function we have

$$b_1 = \frac{\sqrt{3}}{6\pi} g^{(1)}\left(\frac{1}{3}, \tau\right), \qquad b_2 = -\frac{\sqrt{3}}{12\pi}\left(g^{(1)}\left(\frac{1}{3}, \tau\right) - g^{(1)}\left(\frac{1}{6}, \tau\right)\right). \qquad (A.2)$$

Then

$$
\begin{aligned}
f_2 &= -6\left(b_1^2 + 6b_1 b_2 - 4b_2^2\right), \\
f_3 &= 36\sqrt{3}\left(b_1^3 - b_1^2 b_2 - 4b_1 b_2^2 + 4b_2^3\right), \\
f_4 &= 324 b_1^4.
\end{aligned}
\tag{A.3}
$$

In terms of the coefficients $g^{(k)}(z,\tau)$ of the Kronecker function we have

$$
\begin{aligned}
f_2 &= \frac{1}{2\pi^2}\left[3g^{(2)}\left(\frac{1}{2},\tau\right) - g^{(2)}\left(\frac{1}{3},\tau\right) + g^{(2)}\left(\frac{1}{6},\tau\right)\right], \\
f_3 &= \frac{1}{4\pi^3}\left[15g^{(3)}\left(\frac{1}{3},\tau\right) - 12g^{(3)}\left(\frac{1}{6},\tau\right)\right], \\
f_4 &= \frac{1}{4\pi^4}\left[-18g^{(4)}(0,\tau) - 27g^{(4)}\left(\frac{1}{3},\tau\right)\right].
\end{aligned}
\tag{A.4}
$$

The $q$-expansions are

$$
\begin{aligned}
f_2 &= -\frac{1}{2} - 8q - 4q^2 - 44q^3 + 4q^4 - 48q^5 - 40q^6 + \mathcal{O}\left(q^7\right), \\
f_3 &= -3\sqrt{3}\left[q - 5q^2 + 9q^3 - 11q^4 + 24q^5 - 45q^6\right] + \mathcal{O}\left(q^7\right), \\
f_4 &= \frac{1}{4} + 6q + 54q^2 + 222q^3 + 438q^4 + 756q^5 + 1998q^6 + \mathcal{O}\left(q^7\right).
\end{aligned}
\tag{A.5}
$$

In addition we have

$$
\begin{aligned}
\frac{\psi_1}{\pi} &= 2\sqrt{3}(b_1 + b_2) \\
&= \frac{2}{3}\sqrt{3}\left[1 + 3q + 3q^2 + 3q^3 + 3q^4 + 3q^6\right] + \mathcal{O}\left(q^7\right).
\end{aligned}
\tag{A.6}
$$

We define the Eisenstein series $e_2$ by

$$
e_2(\tau) = \frac{1}{2(2\pi i)^2} \sideset{}{'}\sum_{(n_1,n_2)\in\mathbb{Z}^2\setminus(0,0)} \frac{1}{(n_1 + n_2\tau)^2}.
\tag{A.7}
$$

The prime at the summation sign denotes the Eisenstein summation prescription defined by

$$
\sideset{}{'}\sum_{(n_1,n_2)\in\mathbb{Z}^2} f(z + n_1 + n_2\tau) = \lim_{N_2\to\infty}\sum_{n_2=-N_2}^{N_2}\left(\lim_{N_1\to\infty}\sum_{n_1=-N_1}^{N_1} f(z + n_1 + n_2\tau)\right).
\tag{A.8}
$$

The $q$-expansion of $e_2$ starts with

$$
e_2(\tau) = -\frac{1}{24} + q + 3q^2 + 4q^3 + 7q^4 + 6q^5 + 12q^6 + \mathcal{O}\left(q^7\right).
\tag{A.9}
$$

The Eisenstein series $e_2$ is a quasi-modular form.

# B  Boundary values

In this appendix we give the boundary values $B_k$. These are easily obtained from [10,33]. We have

$$
\sum_{k=0}^{\infty} \varepsilon^k B_k = \frac{3}{4} 3^{-\varepsilon}\left[h - \frac{2\pi\varepsilon}{3}\frac{\Gamma(1+2\varepsilon)}{\Gamma(1+\varepsilon)^2}\right],
\tag{B.1}
$$

where

$$h = \frac{1}{i}\left[(-r_3)^{-\varepsilon}\,{}_2F_1\left(-2\varepsilon,-\varepsilon;1-\varepsilon;r_3\right) - \left(-r_3^{-1}\right)^{-\varepsilon}\,{}_2F_1\left(-2\varepsilon,-\varepsilon;1-\varepsilon;r_3^{-1}\right)\right], \qquad \text{(B.2)}$$

and $r_3 = \exp(2\pi i/3)$. The hypergeometric function can be expanded systematically in $\varepsilon$ with the methods of [34–37]. The first few terms are given by

$$\begin{aligned}
{}_2F_1\left(-2\varepsilon,-\varepsilon;1-\varepsilon;x\right) = 1 &+ 2\varepsilon^2 \mathrm{Li}_2(x) + \varepsilon^3\left[2\mathrm{Li}_3(x) - 4\mathrm{Li}_{21}(x,1)\right] \\
&+ \varepsilon^4\left[2\mathrm{Li}_4(x) - 4\mathrm{Li}_{31}(x,1) + 8\mathrm{Li}_{211}(x,1,1)\right] + \mathcal{O}\left(\varepsilon^5\right).
\end{aligned} \qquad \text{(B.3)}$$

The first few boundary values are given by

$$\begin{aligned}
B_0 &= 0\,, \\
B_1 &= 0\,, \\
B_2 &= \frac{3}{2i}\left[\mathrm{Li}_2(r_3) - \mathrm{Li}_2\left(r_3^{-1}\right)\right], \\
B_3 &= \frac{3}{2i}\left\{-2\mathrm{Li}_{21}(r_3,1) - \mathrm{Li}_3(r_3) + 2\mathrm{Li}_{21}\left(r_3^{-1},1\right) + \mathrm{Li}_3\left(r_3^{-1}\right)\right\} - \ln(3)B_2\,, \\
B_4 &= \frac{3}{2i}\left\{4\mathrm{Li}_{211}(r_3,1,1) - 2\mathrm{Li}_{31}(r_3,1) + \mathrm{Li}_4(r_3) - 4\mathrm{Li}_{211}\left(r_3^{-1},1,1\right)\right. \\
&\qquad \left. + 2\mathrm{Li}_{31}\left(r_3^{-1},1\right) - \mathrm{Li}_4\left(r_3^{-1}\right)\right\} - \ln(3)B_3 - \frac{1}{2}\ln^2(3)B_2 + \frac{1}{3}\zeta_2 B_2\,.
\end{aligned} \qquad \text{(B.4)}$$

These can be reduced to polylogarithms of depth 1 as follows [38, 39]:

$$\begin{aligned}
B_2 &= 3\,\mathrm{Im}\,\mathrm{Li}_2(r_3)\,, \\
B_3 &= \frac{24}{5}\mathrm{Im}\,\mathrm{Li}_3\left(\frac{i}{\sqrt{3}}\right) - \frac{17}{90}\pi^3 - \frac{1}{10}\pi(\ln(3))^2\,, \\
B_4 &= -\frac{63}{10}\mathrm{Im}\,\mathrm{Li}_4(r_3) + \frac{48}{5}\mathrm{Im}\,\mathrm{Li}_4\left(\frac{i}{\sqrt{3}}\right) + \frac{17}{90}\pi^3\ln(3) + \frac{1}{30}\pi(\ln(3))^3\,.
\end{aligned} \qquad \text{(B.5)}$$

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
