# Peer review of "On $\varepsilon$-factorised bases and pure Feynman integrals"

_SciPost Physics, doi:SciPost Phys. 16, 150 (2024)_

## Round 2 · Author Response

We implemented all suggestions of the two referees in the revised version, they were helpful.
Only point 2 of report 1 deserves a comment: We added a few lines at the end of section 2 to stress the relevant points from this toy example. The referee asks, if there is a kinematic-dependent redefinition of the master integrals in the polylogarithmic case, which preserves the epsilon-factorised form. One possibility is to multiply a master integral by exp(epsilon*ln(x)), this will do the job. Of course, this transformation is not rational. We felt that specifying in detail what transformations we would allow in this case would divert too much from our main topic.
Only point 2 of report 1 deserves a comment: We added a few lines at the end of section 2 to stress the relevant points from this toy example. The referee asks, if there is a kinematic-dependent redefinition of the master integrals in the polylogarithmic case, which preserves the epsilon-factorised form. One possibility is to multiply a master integral by exp(epsilon*ln(x)), this will do the job. Of course, this transformation is not rational. We felt that specifying in detail what transformations we would allow in this case would divert too much from our main topic.

---

## Round 2 · List of Changes

The requested changes by the referees have been detailed and specific, we followed them point-by-point.
Following remark 1 of referee 1 we revised the complete manuscript with respect to the use of "uniform weight" and "purity".
At the end of section 2 we inserted (point 2 of referee 1):
"This statement is of course obvious to experts in the field.
We will use it in the following way:
If we assume that a definition of transcendental weight beyond the polylogarihmic case
is compatible with the restriction of the kinematic space to a sub-space,
we may detect master integrals of non-uniform weight from their (non-uniform weight) boundary constants
at a point where the master integrals reduce to values of multiple polylogarithms.
For multiple polylogarithms
the definition of transcendental weight is unambiguous."
Point 3 of referee 1: $P(u)$ is now a quartic polynomial of a single variable.
Point 4 of referee 1 and point 4 of referee 2: We added the definition of the complete elliptic integrals.
Point 5 of referee 1: We added the requested definitions and references to standard textbooks.
Point 6 of referee 1: Indeed, and we reformulated this sentence.
Point 7 of referee 1: We added a footnote to explain that this is a useful convention and no restriction of generality.
Point 8 of referee1: We added a footnote how the basis can be found and added references.
Point 1-3 of referee 2: We implemented these changes.
Following remark 1 of referee 1 we revised the complete manuscript with respect to the use of "uniform weight" and "purity".
At the end of section 2 we inserted (point 2 of referee 1):
"This statement is of course obvious to experts in the field.
We will use it in the following way:
If we assume that a definition of transcendental weight beyond the polylogarihmic case
is compatible with the restriction of the kinematic space to a sub-space,
we may detect master integrals of non-uniform weight from their (non-uniform weight) boundary constants
at a point where the master integrals reduce to values of multiple polylogarithms.
For multiple polylogarithms
the definition of transcendental weight is unambiguous."
Point 3 of referee 1: $P(u)$ is now a quartic polynomial of a single variable.
Point 4 of referee 1 and point 4 of referee 2: We added the definition of the complete elliptic integrals.
Point 5 of referee 1: We added the requested definitions and references to standard textbooks.
Point 6 of referee 1: Indeed, and we reformulated this sentence.
Point 7 of referee 1: We added a footnote to explain that this is a useful convention and no restriction of generality.
Point 8 of referee1: We added a footnote how the basis can be found and added references.
Point 1-3 of referee 2: We implemented these changes.

---

## Editorial Decision

published